# Nutrients in the Prevention of Osteoporosis in Patients with Inflammatory Bowel Diseases

**DOI:** 10.3390/nu12061702

**Published:** 2020-06-06

**Authors:** Alicja Ewa Ratajczak, Anna Maria Rychter, Agnieszka Zawada, Agnieszka Dobrowolska, Iwona Krela-Kaźmierczak

**Affiliations:** Department of Gastroenterology, Dietetics and Internal Diseases, Poznan University of Medical Sciences, 49 Przybyszewskiego Street, 60-355 Poznan, Poland; a.m.rychter@gmail.com (A.M.R.); aga.zawada@gmail.com (A.Z.); agdob@ump.edu.pl (A.D.)

**Keywords:** bowel diseases, diet, osteoporosis, bone density, nutrients

## Abstract

The chronic character of inflammatory bowel diseases, such as Crohn’s disease and ulcerative colitis, results in various complications. One of them is osteoporosis, manifested by low bone mineral density, which leads to an increased risk of fractures. The aetiology of low bone mineral density is multifactorial and includes both diet and nutritional status. Calcium and vitamin D are the most often discussed nutrients with regard to bone mineral density. Moreover, vitamins A, K, C, B12; folic acid; calcium; phosphorus; magnesium; sodium; zinc; copper; and selenium are also involved in the formation of bone mass. Patients suffering from inflammatory bowel diseases frequently consume inadequate amounts of the aforementioned minerals and vitamins or their absorption is disturbed, resulting innutritional deficiency and an increased risk of osteoporosis. Thus, nutritional guidelines for inflammatory bowel disease patients should comprise information concerning the prevention of osteoporosis.

## 1. Introduction

Inflammatory bowel diseases (IBD), including Crohn’s Disease (CD) and Ulcerative Colitis (UC), are chronic conditions, the aetiology of which is not entirely known. Possible risk factors comprise genetic predisposition, immunological disorders, and environmental conditions [1]. Furthermore, the chronic character of these diseases causes extra-intestinal complications such as osteoporosis, which is manifested by low bone mineral density, resulting in an increased risk of fractures [2].

The World Health Organisation (WHO) classifies Bone Mineral Density (BMD) on the basis of Dual-Energy X-ray Absorptiometry (DEXA)—i.e., the gold standard in the diagnosis of osteoporosis:Normal: T-score ≥ −1 SD;Osteopenia (low bone mass): T-score < −1 SD and >−2.5 SD;Osteoporosis: T-score ≤ −2.5 SD;Severe osteoporosis: T-score ≤ −2.5 SD with fragility fractures [3].

Risk factors of osteoporosis in IBD include early age onset; steroid therapy; malnutrition; low body mass; hormonal disorders, including a decreased oestrogen level; and malabsorption causing a nutritional deficiency, particularly in terms of vitamin D and calcium. However, osteoporosis may also stem from genetic factors. In the course of IBD, pro-inflammatory molecules TNF-α, IL-1β, IL-6, and IL-17 levels are elevated and lead to increased bone resorption, causing a decrease in bone mineral density [4]. In fact, low bone mineral density, osteopenia and osteoporosis were found in 22–77%, 32–36%, and 7–15% of IBD patients, respectively [1]. Additionally, low BMD is observed more frequently in CD than UC patients [5]. 

An important risk factor for BMD loss in IBD patients is the use of certain medications. Corticosteroids (CS) are a group of medications whose prolonged use is associated with side effects involving bone tissue [5]. CS increase apoptosis and decrease the formation of osteoblasts and promote osteoclastogenesis. Moreover, CS influence calcium balance (decreasing calcium absorption in the intestine and renal resorption) and the neuroendocrine system. The data indicate that following the first year of steroid therapy, bone mass may decrease by about 12%, and 2–3% per year in the following year. In addition, CS use decreases muscle mass, which leads to an elevated risk of fractures [6,7]. Corticosteroids also affect the receptor activator of nuclear factor κβ/receptor activator of nuclear factor κβ ligand/osteoprotegerin (RANK/RANKL/OPG) pathway. In fact, CS increase the expression of RANKL, which binds with the RANK receptor in osteoclasts, which in turn elevates their differentiation and activation. Moreover, corticosteroids decrease the level of OPG responsible for the inhibition of RANKL activity [8]. According to research studies, therapy with steroids was associated with a lower BMD in IBD patients [9]. Nevertheless, different results obtained in other studies may stem from differences in the methodology of the research. Wada et al. reported that steroid therapy was a risk factor for low BMD for UC patients but not for CD patients [10]. In our study, we also demonstrated a correlation between a cumulative prednisolone dose, administered in the course of the disease, and the lumbar spine (L2–L4) T-score, the femoral neck (FN) T-score, and the Z-score in UC patients. However, the above-mentioned association was not found in CD patients [5]. It is generally accepted that osteoporosis affects 30–50% of subjects treated with CS. Therefore, in order to prevent steroid-related osteoporosis, the BMD should be evaluated before and after every year of CS treatment. Additionally, the supplementation of calcium (1200 mg/day) and vitamin D (800 IU) should also be recommended [11].

As far as Infliximab (IFX) is concerned, research studies suggest that IFX treatment in CD patients increased the N-terminal telopeptide of type I collagen [12]. However, another study showed no difference in the level of N-terminal telopeptide in type I collagen, although it was pointed out that the bone-specific alkaline phosphatase and total osteocalcin were elevated in CD patients who had used IFX [13]. The BMD and bone mineral content of the lumbar spine increased in CD patients treated by IFX [14]. Lima et al. reported the association between azathioprine and infliximab use and low BMD in a patient with CD [15]. However, another study revealed that IFX did not affect BMD [16]. Krajcovicova et al. showed that a combined therapy with anti-TNFα and azathioprine improved the lumbar spine BMD by –3% per year, whereas corticosteroid therapy reduced the BMD by –3% per year [17]. Moreover, Hoffmann et al. reported that immunomodulatory treatment was associated with a BMD decrease [18], although another study revealed that biological and immunomodulatory treatment did not affect the risk of hip fracture [19].

The use of monoclonal antibodies in osteoporosis associated with IBD is the subject of several studies. Bone metabolism at the osteoclast/osteoblast level is regulated by the following factors belonging to the tumour necrosis factor superfamily: receptor activator of nuclear factor kappa B (κB; RANK), its ligand (receptor activator of nuclear factor κB ligand, RANKL), and osteoprotegerin (RANKL/RANK/OPG). Krela-Kaźmierczak I. et al. showed that low OPG levels may be associated with osteoporosis in UC. Nevertheless, in CD patients increased RANKL levels were observed [20,21]. Interleukin 6 can modulate BMD via the OPG/sRANKL system in the femoral neck, which can cause a loss of bone, particularly in the course of CD. Therefore, these studies may contribute to the development and implementation of new secondary osteoporosis therapy in the course of IBD [22]. It is worth emphasizing the effects of biological therapy employed in IBD on bones. Although studies focusing on the effects of anti-TNF-α therapy on bone metabolism in IBD are limited, it was observed that IFX treatment induced beneficial effects on bone metabolism in IBD patients [23,24], which suggests that monoclonal antibodies constitute an important factor in the treatment of osteoporosis. Understanding the role of RANKL and sclerostin in bone cell biology entirely changes the therapeutic perspective. Sclerostin, an antagonist of the Wnt pathway, plays a key role in bone formation and is mainly secreted by osteocytes. In fact, high levels of RANKL and sclerostin were detected in osteoporosis, leading to the production of antibodies which are able to neutralize their activity. Denosumab (anti-RANKL antibody) is a fully human monoclonal antibody which inhibits osteoclastic-medicated bone resorption by binding to osteoblast-produced RANKL. Romosozumab, a monoclonal antibody binding sclerostin, increases bone formation and decreases bone resorption. Denosumab and romosozumab present promising results in the treatment of postmenopausal osteoporosis [25]. Therefore, the use of these antibodies in the therapy of secondary osteoporosis in IBD patients requires further research.

Diet and nutritional status constitute risk factors in the development of osteoporosis. Micro and macronutrients, vitamins, and mineral components intake is often below the recommended values, particularly when the disease is active. Hence, malnutrition and deficiency in nutrients essential to bone mineralization can affect patients with inflammatory bowel diseases [26].

## 2. Proteins, Fats, and Carbohydrates Intake in IBD and the Risk of Osteoporosis Development

The European Society for Clinical Nutrition and Metabolism (ESPEN) recommends supplying 1.2–1.5 g protein/kg body weight in active IBD. Patients in remission should consume 1 g protein/kg body weight. Improper diet and the loss of nutrients in the gastrointestinal tract leads to an increase in fat mass, as well as a decrease in free fat mass in patients suffering from IBD. On the basis of these guidelines, there is no special diet in the remission period [27]. The recommended macronutrient intake values for adults are listed below (Table 1) [28,29].

### 2.1. Protein

Protein provides about 50% of bone volume, 1/3 of bone mass and forms the bone matrix, therefore its supply determines the maintenance of bone mass in adults [30]. On the other hand, an excessive intake of protein causes increased calcium excretion and negative calcium balance [31].

Moderate or severe hypoalbuminemia was diagnosed in 17% and 24% of IBD patients, respectively. In patients with albumin deficiency following surgical procedures, mortality within 30 days was higher than in patients with normal albumin levels [32]. Protein-energy wasting is observed more frequently in IBD patients than in healthy individuals, and the same observation can be made in Crohn’s disease patients as compared to subjects suffering from ulcerative colitis [33]. There were no statistically significant differences between men with ulcerative colitis in remission and healthy subjects [34].

One of the research studies showed a statistically significant difference in protein intake between men over 63 years old with and without a femur fracture. Every 3% increase in protein daily intake caused a 20% decreased risk of femur fractures [35]. What is more, the percentage of protein in the daily caloric content of middle aged women (65–72 years old) correlated negatively with a femoral neck. Additionally, a protein intake higher than 1.2 g/kg body weight was associated with lower BMD in the femoral neck and lumbar spine [36].

### 2.2. Fats

There was no statistically significant differences in the intake of total fat, saturated fat, and mono–and poly–unsaturated fat between healthy individuals and CU patients [34]. The European Society for Clinical Nutrition and Metabolism does not recommend n-3 fatty acid supplementation for maintaining remission in IBD [27].

The development of osteoporosis was correlated with the type of fatty acids. In fact, the risk of fractures was increased by n-3 and saturated fatty acid intake and it was decreased by monounsaturated and n-6 fatty acids. Dong et al. demonstrated that the supplementation of eicosapentaenoic and docosahexaenoic acid (1.2 g per day) did not result in statistically significant differences in osteocalcin and bone-specific alkaline phosphatase [37]. Furthermore, n-3 fatty acid supplementation did not significantly alter bone-specific alkaline phosphatase and type I collagen-telopeptide levels, although it decreased the osteocalcin level [38]. The twenty-four-week supplementation of n-3 fatty acid in combination with aerobic exercise reduced chronic inflammation and increased BMD in postmenopausal women [39].

### 2.3. Carbohydrates

Carbohydrates supply energy for host cells and the intestinal microbiome. Fermentable carbohydrates (especially mono–and di–saccharides) and fibre constitute important factors in inflammatory bowel diseases [40].

A meta-analysis demonstrated that a low-FODMAP diet (low-Fermentable Oligosaccharides, Disaccharides, Monosaccharides, and Polyol) decreased the incidence of abdominal pain, nausea, weakness, and flatulence, although no change in the rate of constipation was observed [41]. Patients with IBD in remission were supplied with insufficient amounts of carbohydrates relative to recommendations [42]. “National Health and Nutrition Examination Survey” research demonstrated a negative association between energy density (a diet rich in carbohydrates, sugar, fat, and saturated fatty acid) and BMD in the femur and femoral neck [43]. In animals, a high fructose diet decreased transepithelial active calcium ion transport and the 1.25(OH)2D3 level by about 30–40% [44].

On the other hand, the fructose content in breast milk triggered an increased BMC (Bone Mineral Content) in a 6-month-old infant compared to the first month of the baby’s life. Nevertheless, lactose and glucose did not cause changes in BMC [45]. Another study pointed to the association between carbonated beverages consumption, which frequently are sources of sugar, and fractures [46]. In fact, a meta-analysis showed that the consumption of non-alcoholic beverages was negatively associated with calcium supply [47]. Additionally, fibre possibly prevents a decrease in BMD, as water-soluble fibre intake causes increased calcium retention in bones [48].

## 3. The Importance of Vitamin Intake in IBD and Bone Mineral Density

### 3.1. Vitamin D

Vitamin D comprises a group of chemical compounds such as ergocalciferol (vitamin D2) and cholecalciferol (vitamin D3). The active forms of vitamin D are 1.25(OH)2D and 24.25(OH)2D. Vitamin D acts directly and indirectly on the bone [49], regulates calcium absorption in the gastrointestinal tract, determines the proper serum calcium level, influences osteoblasts mineralization and differentiation, and decreases the synthesis of parathormone and the reabsorption of phosphates from bones. Vitamin D can be found in fatty sea fish, cod liver oil, and yolk [49,50]. 

IBD patients are a group presenting a higher risk of vitamin deficiency and, therefore, should control their 25(OH)D serum level. The recommended vitamin D supplementation for healthy adults is 800–2000 IU from September to April [51]. According to the available data, vitamin D serum level was lower than 30 ng/mL in IBD, but without statistically significant differences compared to the control group. It may stem from the low exposure to sunlight of the whole population and malabsorption caused by the inflammation in the gastrointestinal tract associated with IBD [52]. The deficiency was observed in 52% of children with IBD [53]. A sufficient vitamin D level was found more often in patients with a non-inflammatory disease activity; however, the serum concentration level was not correlated with any inflammatory markers [54]. In terms of IBD patients, a vitamin D dose of 1820 IU was sufficient to maintain the normal 25(OH)D serum level [55]. 

Although the supplementation of vitamin D decreased the risk of fractures in general as well as the risk of femoral neck fractures, it did not alter the risk of osteoporosis [56]. In elderly patients, the 1-year supplementation of vitamin D (12,000, 24,000, or 48,000 IU per month) did not change the hip BMD but decreased the parathormone level [57]. A meta-analysis showed the supplementation of vitamin D in patients over 50 years of age was associated with an increased risk of fractures. The dose of 1200 mg calcium and minimum 800 IU of vitamin D was more effective than lower doses [58]. However, a high dose (4000 IU and more) of vitamin D decreased the volumetric BMD in healthy adults. The study did not reveal significant differences in bone strength [59].

### 3.2. Vitamin C (Ascorbic Acid)

Bone tissue is 90% made of collagen, and vitamin C is a necessary antioxidant for its synthesis [60]. Moreover, it is possible that ascorbic acid participates in the synthesis of osteoblasts and influences osteoclast differentiation [61]. In animals, vitamin C deficiency increases osteoclast and decreases bone formation [62].

Since patients with IBD fear the symptoms of the disease, they often eliminate vegetables and fruits from their diets. This, in turn, may lead to an insufficient vitamin C intake and to the deterioration of the patients’ health. Children, the elderly, and patients with the malabsorption syndrome are particularly exposed to this deficiency. Ascorbic acid can be found in citrus fruits, potatoes, parsley, and berries without seeds; thus, their consumption should be encouraged in this patient group [63]. In fact, patients with inflammatory bowel diseases consumed significantly less fruits and fewer vegetables; hence, they presented lower serum vitamin C than healthy individuals [64]. Patients suffering from ulcerative colitis consumed a similar amount of vitamin C in comparison with the healthy subjects; however, the INQ (Index of Nutritional Quality) of UC patients was significantly lower than that of healthy adults [65]. What is more, children with IBD presented a significantly lower intake of vitamin C than healthy children [66]. Enteral nutrition decreased the serum vitamin C level in children with active Crohn’s disease, whereas glutamine supplementation did not influence the ascorbic acid level [67].

Sahni et al. estimated vitamin C intake on the basis of the Food Frequency Questionnaire. Following a 17-year-long observation period, subjects who consumed the highest amount of vitamin C suffered femoral neck and non-spine fractures less frequently than individuals reporting the lowest intake. The authors suggested that vitamin C intake in the elderly can be beneficial to bone health [60]. Additionally, daily vitamin C intake was associated with BMD in postmenopausal women. Patients with osteoporosis consumed a smaller amount of ascorbic acid than women not suffering from osteoporosis [68]. On the other hand, an American study presented a lack of association between vitamin C intake (from food and supplements) and BMD in postmenopausal women. In fact, long-term vitamin C supplementation was connected with a higher BMD in 55 to 64-year-old women [69]. Furthermore, although it was just a pilot study comprising 34 subjects, it showed that anti-oxidative vitamins influenced BMD, vitamin E (600 mg), and vitamin C (1000 mg) with and without resistance training, as well as preventing a decrease in BMD [70]. Kim et al. reported that a higher vitamin C intake was associated with a lower risk of osteoporosis development in subjects over 50 years of age; however, this was only in patients reporting little physical activity [71].

### 3.3. Vitamin B12 and Folic Acid

Vitamins B9 (folic acid) and B12 (cobalamin) are water-soluble vitamins. Folic acid is necessary for DNA methylation, whereas cobalamin is a coenzyme found in many biochemical reactions in the human body, e.g., in the metabolization of folic acid, or DNA synthesis. Their deficiency causes megaloblastic anaemia as well as nervous system disorders [72]. Vitamin B12 and folic acid are involved in the remethylation of homocysteine to methionine; therefore, a deficiency in these vitamins leads to hyperhomocysteinemia. It is vital to bear in mind that the serum homocysteine level is correlated to osteoporotic fractures [73]. A deficiency in folic acid and vitamin B12 is due to insufficient intake, abnormal absorption, or increasing demand. Furthermore, cobalamin connects with an intrinsic factor, allowing for the absorption of vitamins in the ileum, and folic acid is absorbed in the duodenum and jejunum. Both of these vitamins are not absorbed in the large intestine. The risk of B9 or B12 deficiency is lower in ulcerative colitis than in Crohn’s disease, which may stem from the fact that CD can affect all parts of the gastrointestinal tract, including the small intestine, whereas inflammation in UC predominantly affects the large intestine [74].

A meta-analysis showed that the serum folic acid level was lower in IBD patients than in healthy individuals. This could be accounted for by an insufficient intake, higher utilization, or taking medications. The authors recommended the routine supplementation of folic acid for colorectal cancer prevention. However, the study did not demonstrate a significant difference in serum B12 levels between the patients and the control group [72].

Kim et al. pointed out a negative correlation between the serum homocysteine level and BMD in women under 50 years of age [75]. Women with a normal homocysteine concentration consumed a larger amount of vitamin B12 than women with a higher level. In the same group, subjects with a normal lumbar spine BMD presented a lower serum homocysteine level and higher folic acid concentration in erythrocyte than women with low BMD or osteoporosis [76]. There was no correlation between the serum vitamin B12 level and the risk of fractures in women and men. However, an association was found between homocysteine concentration and the risk of a femoral neck fracture, particularly in women [77]. Salari et al. investigated the influence of folic acid supplementation (1 mg/day) on the serum osteocalcin and homocysteine level in postmenopausal women. They observed a significant difference in osteocalcin concentration between the study and control group after 6 months. It was found that the serum homocysteine level decreased in both groups, although the changes were not significant within the groups or between them [78]. Vitamin B9 and B12 supplementation was demonstrated to decrease homocysteine concentration and increase serum folic acid and cobalamin levels. Serum alkaline phosphatase and C-terminal cross-linking collagen I telopeptide levels did not change significantly [79].

Hyperhomocysteinemia (diagnosed in 60% of patients) was associated with osteoporosis and low BMD in CD patients, whereas bone disorders affected 90% of the subjects [80].

### 3.4. Vitamin A

Vitamin A is vital for growth; vision; and osteoclastogenesis stimulation, which further influences BMD. A total of 16% of non-adult patients with IBD presented vitamin A deficiency [81].

There was no statistically significant difference in vitamin A intake between men suffering form UC and the control group [34]. In fact, vitamin A or beta-carotene supplementation within a 15-year period did not increase the risk of bone fractures [82]. The retinol serum level correlated negatively with the BMD of the lumbar spine and femoral neck in postmenopausal women [83]. A meta-analysis showed that a high vitamin A level but not beta-carotene intake increased the risk of a femoral neck fracture. Additionally, a high and a low serum vitamin A level increased the risk of a femoral neck fracture. The authors did not observe a significant difference in total fractures depending on the supply and vitamin A concentration [84].

### 3.5. Vitamin K

The enzyme participating in the synthesis of glutamic acid (Gla) found in osteocalcin created in osteoblast and in the matrix bone protein is vitamin K. It occurs in two forms: K1 (phylloquinone) and K2 (menaquinone, MK-n) [85].

The study, conducted among postmenopausal Japanese women with osteoporosis, indicated that the administration of risedronic acid with vitamin K (45 g/day) was not more beneficial than administration without vitamin K supplementation. In fact, there was no statistically significant difference in fractures within the studied group [86]. Another study demonstrated that supplementation of menaquinone decreased carboxylated osteocalcin to a undercarboxylated osteocalcin ratio in young adults (47 ± 14 years old) and in postmenopausal women [87]. However, the supplementation of 360 ng of vitamin K (MK-7) for 12 months did not change bone mass as compared to the control group. The authors observed an increase in the serum carboxylated osteocalcin and a decrease in the undercarboxylated osteocalcin level in the study group in comparison with the placebo group [88].

## 4. Minerals and Bone Mineral Density in IBD

### 4.1. Calcium and Phosphate 

Calcium (Ca) is responsible for proper BMD, blood coagulation, and the proper functioning of the cardiovascular system. In the human body, more than 99% of Ca is stored in bones. Therefore, a decreased serum calcium level leads to its release from bones and causes bone tissue resorption [50]. Furthermore, an insufficient calcium intake causes hormonal disorders, leading to a higher risk of fractures. Calcium can be found in such sources as milk, dairy products, and green leafy vegetables [89]. Additionally, the human body contains about 700 g of phosphorus (P), which is mainly stored in bones (80–90%). Hence, both its excessive and inadequate intake can develop osteoporosis. Phosphorus deficiency, or its insufficient supply to calcium supply ratio, causes bone resorption and inhibits bone mineralization and bone formation. On the other hand, an oversupply of P, particularly with insufficient Ca intake, results in excessive parathormone excretion and the loss of bone mass [90].

The insufficient intake of calcium was estimated in 80–86% of IBD patients, who avoid milk and dairy products due to lactose intolerance [91]. Patients with IBD have lower calcium and phosphate levels in comparison with healthy individuals [52].

Another cause of calcium malabsorption is the use of steroids as well as the occurrence of diarrhoea. The supplementation of calcium in a 1000–1500 mg/day dose is recommended for most patients with inflammatory bowel diseases. Furthermore, patients treated with steroids require calcium and vitamin D supplementation [91]. Calcium intake was correlated negatively with the femoral neck BMD but not with the lumbar spine BMD in IBD patients [92]. Premenopausal women suffering from IBD consumed insufficient amounts of calcium and vitamin D, and their intake of Ca and vitamin D was correlated [93]. Moreover, a low calcium serum level was observed in patients more frequently than in the control group, although it was insignificant. Additionally, the Ca serum level was negatively correlated with steroids [52]. A meta-analysis demonstrated that calcium supplementation without other substances (for example, vitamin D) did not alter the risk of femoral neck fractures in both sexes [94]. The study revealed that an increased intake of calcium by every 300 mg decreased the risk of fractures, although it was nonlinear. The highest risk was found in the intake below 751 mg of calcium. The fracture risk was unchanged in the intakes of more than 1137 mg and 882–996 mg of calcium [56]. Gutiérrez et al. demonstrated that a one-week diet rich in phosphorus (1677 ± 167 mg/day) increased Fibroblast Growth Factor 23 (FGF23), osteocalcin, and osteopontin levels. The aforementioned results suggest that a phosphorus-rich diet negatively affects health [95], and that women over 45 years of age, both with and without osteoporosis, consume similar amounts of calcium. Thus, Ca intake was not associated with the incidence of fractures [96].

### 4.2. Magnesium

Magnesium (Mg) is absorbed in the small intestine, and its absorption ranges from 30% to 80%. Bones store about 60% of the total body magnesium. The main sources of Mg are legumes, seeds, nuts, almonds, spinach, and buckwheat. Not only is this element responsible for the stability and permeability of cell membranes but it also maintains the DNA double helix integrity and regulates the activity of about 300 enzymes [97]. On the other hand, magnesium deficiency causes decreased osteoblast and osteoclast activity, resulting in bone metabolism disorders [98]. Chronic hypomagnesemia leads to the disturbance of parathyroid hormone production, leading to hypocalcaemia [99].

Patients with UC and CD consumed a lower amount of Mg than healthy adults. CD patients consumed 60–63% of the daily magnesium requirement [65,100]. Magnesium intake correlated with BMD, with a stronger correlation found in men than in women [98]. Postmenopausal women who consumed 422.5 mg and more of Mg per day presented a higher hip and total body BMD by 3% and 2%, respectively, than the individuals supplying <206.5 mg Mg/day. No association was observed between magnesium intake and the risk of fractures. On the other hand, a high magnesium dose was associated with a higher risk of forearm and wrist fractures in comparison with a low Mg intake. The authors paid attention to the subjects with a high supply of magnesium, since they reported much physical activity, which increases the frequency of falls [101]. The supplementation of magnesium (106 mg) and calcium (1200 mg) for 4 weeks in postmenopausal women did not change the serum parathyroid hormone level both in the study and the control group. However, the supplementation increased the serum CTX (C-terminal telopeptide) level—i.e., a bone resorption marker [102]. A conducted meta-analysis indicated that a high magnesium intake was not associated with a lower risk of hip fractures. On the other hand, magnesium dose was connected with the hip and femoral neck BMD, although no association was found with the lumbar spine BMD [103].

### 4.3. Sodium (Na)

The absorption of water and electrolytes, including sodium (Na), takes place in the colon. The lymphatic function of the large intestine can be impaired in the course of the mucosal inflammation [104].

In spite of the fact that the human body contains as much as 105 g of sodium, the intake of Na in the population is still too high, with some people consuming 9–12 g salt per day, which results in numerous disorders and also affects bones [105]. Na is known to improve taste and preserve products [106]; it is usually found in salt (40% of mass), meat and its preparations, grains, milk, and dairy products. This element constitutes the main extracellular cation excreted in urine and sweat. Moreover, sodium is responsible for the maintenance of the acid-base balance, cell work, and the transmission of nerve impulses. Although a normal sodium Na^+^ level is 135–145 mmol, both too high and too low Na concentration levels constitute a threat to health and life. In fact, hypernatremia causes weakness, headache, vomiting, loss of appetite, weak nerve reflexes, and cardiac disorders. On the other hand, hyponatremia induces neuromuscular excitability, confusion, and cardiac arrest.

Patients with UC in remission consumed non-significantly lower sodium amounts than healthy individuals [34]. The sodium intake was lower in malnourished subjects than in properly nourished patients [107].

The Korea National Health and Nutrition Examination Survey indicated that osteoporosis was observed more frequently in postmenopausal women consuming ≥ 4001 mg of salt per day than in those consuming ≤ 2000 mg/day. A salt intake of ≥ 5001 mg was associated with a higher risk of osteoporosis in the femoral neck compared to the consumption of ≤ 2000 mg/day [108]. A sodium-rich diet (11.2 g of salt per day) increased calcium excretion in urine and changed the serum NTX (N-terminal telopeptide) level in comparison with a low salt intake (3.9 g). However, there was no significant change in the concentration of Pyr (pyridoxine) and Dpyr (deoxypyridoline) [109]. A meta-analysis demonstrated that a high intake of Na is a factor associated with a higher risk of osteoporosis. There was no significant correlation between the amount of calcium excretion in urine and bone mineral density [110].

A low sodium diet (2 g salt/day) for 6 months decreased calcium excretion with urine in patients who consumed 3.4 g or more salt per day and reduced the concentration of P1NP (propeptide of type 1 collagen). There was no significant change in the serum NTX level.

On the other hand, a low-sodium diet (2 g salt per day) of 6-month duration in persons consuming 3.4 g or more salt per day increased the amount of excreted calcium and the serum P1NP level. The authors did not observe any changes in the serum NTX level [111].

## 5. Microelements and Bone Mineral Density in IBD

### 5.1. Zinc

Zinc (Zn) catalyses about 100 enzymes, which makes it an indispensable element for the proper functioning of the immune system as well as for protein and DNA synthesis [112]. Zinc participates in the stabilization of the mucous membrane structure and inhibits mast cells degranulation. On the other hand, zinc deficiency can lead to the release of endogenous heparin, which contributes to the development of osteoporosis. Nevertheless, the exact mechanism of zinc influence on the prevention of osteoporosis is unknown [113].

About 15% of patients with UC or CD exhibited zinc deficiency. In diarrhoea, which often affects IBD patients, Zn demand in the body increases, and its supplementation is essential [112].

The serum zinc level of postmenopausal women with osteoporosis was lower than that of the control group, suggesting the influence of Zn on bone mineralization [113]. Zinc intake correlated negatively with the excretion of Pyr and Dpyr, which are specificity and sensitivity bone resorption markers. No association with serum osteocalcin and alkaline phosphatase was observed, and more patients presented a normal serum zinc level [114].

### 5.2. Copper

The human body contains about 80–150 mg copper (Cu), stored mainly in the muscles, bones, brain, and liver. Cu is absorbed in the small intestine and constitutes a cofactor for many enzymes, such as lysyl oxidase, participating also in collagen synthesis [115]. Thus, copper deficiency results in anaemia, the rupture of blood vessels, fractures, and depression [116]. 

Although women with IBD did not consume enough copper [42], Nangliyai et al. demonstrated that the copper serum level was statistically higher in IBD patients than in the control group. In fact, BMD correlated negatively with the serum Cu level [117].

The serum Cu level was statistically lower in postmenopausal women with osteoporosis than in the group without bone disorders. The administration of calcitonin to women with osteoporosis caused an increase in the serum Cu level. Hence, the results suggested that Cu impacts the development of osteoporosis [112]. In contrast, Cu supplementation (3 or 6 mg per day) for 4 weeks did not cause a change in the level of urine osteocalcin, creatinine, and the daily excretion of Pyr and DPyr [118].

### 5.3. Selenium

Selenium (Se) is an antioxidant and a cofactor for numerous enzymes in the human body. Additionally, it decreases inflammation and inactivates osteoclasts, thus influencing BMD [119]. Furthermore, since deiodinase contains Se, this element determines thyroid function. The main sources of selenium are eggs, meat, and milk, but only if it was added to fodder [120].

It was observed that the serum selenium level was lower in CD patients than in healthy adults. In men with UC, the selenium concentration depended on the location of the inflammatory lesions [121].

A study showed a negative correlation between the amount of selenium intake and osteoporosis [122]. On the other hand, the serum selenoprotein level correlated positively with total BMD, and the serum selenium level with the total and femoral neck BMD in aging men from Europe. It is vital to note that the observed relationships were independent of the thyroid function [123].

The demand for selected vitamins and minerals for adults is presented in Table 2.

## 6. Polyphenols

Polyphenols are bioactivity chemical compounds which occur in plants, primarily in fruits, vegetables, coffee, tea, cocoa, herbs, and spices [124]. They are antioxidants and possess anti-inflammatory properties, which may influence the course of inflammatory diseases, such as IBD [125]. Polyphenols may influence the activity of nuclear factor erythroid 2–related factor 2 (Nrf2), which protects against oxidative stress and inflammation [126]. Moreover, polyphenols affect gut microbiota composition, stimulating bifidium, and lactobacilli [127]. There only a few studies regarding the influence of selected polyphenols on the IBD course. The available research is more frequently focused on animal models.

In addition, it is crucial to acknowledge the importance of green tea polyphenols. They decrease the TNF-α level and increase COX-2 (cyclooxygenase-2) synthesis. [128].

A study based on animal models showed that green tea polyphenols increase antioxidants levels, reduce serum TNF-α and Il-6, and decrease intestinal inflammation. [129]. Rezaei et al. reported that feeding rats with honey (which contains phenols and flavonoids) and Spirulina Platensis (containing antioxidants) may protect against UC development induced by acetic acid. The supplementation of polyphenols influences, for instance, the downregulation of various inflammatory pathways [130].

Another study presented no association between the total polyphenol intake and CD or UC, but polyphenols and resveratrol may protect against the development of Crohn’s disease [124]. Moreover, the 6-week supplementation of resveratrol (500 mg/day) in UC patients decreased the TNF-α level and the clinical colitis activity index score [131].

What is more, polyphenols are vital elements in the prevention of osteoporosis. The consumption of products with high amounts of polyphenols (olive oil, soy, tea, fruits, vegetables) probably protects against osteoporosis development. However, the mechanism of action is not well understood [132]. A higher intake of flavonoids was associated with a higher BMD of the spine, but no association was found with the hip BMD [133]. Hardcastle et al. reported a negative correlation between the amount of flavonoids intake and bone resorption markers [134].

## 7. Conclusions

The diet of IBD patients often proves to be inadequate, which affects the peak bone mass, thus leading to osteopenia, osteoporosis, and fractures. Nutrition guidelines for IBD patients should include osteoporosis prevention.

In conclusion, it is worth emphasizing that:The vitamin D concentration in patients with IBD should be examined routinely, since IBD constitutes a risk factor of vitamin D deficiency. Individual doses of vitamin D are recommended.Most IBD patients require calcium supplementation (1000–1500 mg/day).We recommend the assessment of the fruit and vegetable intake in IBD patients. In patients with low BMD and a simultaneous inadequate consumption of fruits and vegetables, vitamin C supplementation may be considered.Vitamin B12 and folic acid supplementation may be introduced in patients with hyperhomocysteinemia, since a high level of homocysteine correlates with a low bone mineral density.Vitamin A supplementation is not recommended. A high level of this vitamin may have a negative influence on bone health.There are no sufficient data to recommend the supplementation of vitamin K, magnesium, zinc, copper, and selenium in IBD patients for the prevention of osteoporosis.IBD patients consume less sodium than the rest of the population, in particular the malnourished patients. On the other hand, sodium intake in developed countries is higher than recommended, and a high dose of Na correlates with a low BMD. Therefore, it is not necessary to encourage patients to use more salt.Polyphenols may be beneficial for patients’ health; thus, they should be included in the diet in the form of herbs, fruits, or vegetables.

The importance of diet in the prevention of osteoporosis in the group of patients with IBD is indisputable. Hence, it is vital to adhere to the recommendations of a clinical dietitian, as an important member of the coordinated IBD patient care team.

## Figures and Tables

**Table 1 nutrients-12-01702-t001:** References of macronutrient consumption.

Macronutrient	DRI * (% of Total Energy Intake)	RDA **	
		% of Total Energy Intake	Mass of Macronutrient
Protein	10–35	10–20	0.9 g/kg body weight/day
Carbohydrates	45–65	50–70	Minimum 130 g/day
Fat	20–35	20–35	53–158 g/day
n-6 fatty acids (linoleic acid)	5–10	4	-
n-3 fatty acids (alfalinolenic acid)	0.6–1.2	0.5	-

* DRI—Dietary Reference Intake [28]; ** RDA—Recommended Dietary Allowance [29].

**Table 2 nutrients-12-01702-t002:** Requirements for the selected vitamins and minerals in adults [29].

Nutrient	Male	Female
Vitamin A * (μg equivalent of retinol)	900	700
Vitamin D ** (μg cholecalciferol)	15	15
Vitamin K ** (μg phylloquinone)	65	55
Vitamin C * (mg)	90	75
Folic acid * (μg equivalent of folate)	400	400
Vitamin B12 * (μg cobalamin)	2.4	2.4
Calcium * (mg)	1000–1200	1000–1200
Phosphorus * (mg)	700	700
Magnesium *(mg)	400–420	310–320
Sodium ** (mg)	1200–1500	1200–1500
Zinc * (mg)	11	8
Copper * (mg)	0.9	0.9
Selenium * (μg)	55	55

* RDA—Recommended Dietary Allowance; ** AI—Adequate Intake.

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
