# Peer review of "Nutrients in the Prevention of Osteoporosis in Patients with Inflammatory Bowel Diseases"

_nutrients, 2020, doi:10.3390/nu12061702_

Round 1

Reviewer 1 Report

I have started the review of the authors' revised manuscript (see the attached file) but the language editing in this manuscript remains significantly below par, even after the authors hired a translation service. I strongly suggest the authors hire MDPI Author Services for Specialized Editing, which takes up to four business days to complete.

https://www.mdpi.com/authors/english

Reviewer 2 Report

This is a revised manuscript by Ratajczak et al and authors whereby they provide a systematic review about the role of micro and macronutrients in the prevention of osteoporosis in IBD patients. The authors did a commendable job of English improvement and have provided a better version of the manuscript. Previously raised concerns were also addressed and are to the point. I am satisfied with the review and have no further comments to make. 

Minor point:

One of the paragraph is in bold and should be dehighlighted

Reviewer 3 Report

The review is interesting and well written, few items should be addressed:

  • a small paragraph about the role of polyphenols on IBD should be reported;
  • Monoclonal antibodies are important for the cure of osteoporosis (refer to Expert Opin Biol Ther. 2018 Feb;18(2):149-157.); what about their use in osteoporosis associated to IBD?
  • line 54 osteoclast genesis should be corrected;

Round 2

Reviewer 1 Report

Please see my comments in attached pdf file.

Author Response

This manuscript is a resubmission of an earlier submission. The following is a list of the peer review reports and author responses from that submission.

Round 1

Reviewer 1 Report

This manuscript is seriously flawed with unreadable English. See attached file for examples of corrected English. The authors must seek the services of a professional English editing service, and resubmit.

Reviewer 2 Report

The manuscript by Ratajczak et al provides a systematic review about the role of nutrients in the prevention of osteoporosis in IBD patients. The strength of review is that it describes various studies detailing requirement and role of various macro and micronutrients in IBD. However,there are some comments/concern which can make the review stronger

The authors should carry out extensive edinting and grammatically correct English throughout manuscript. As of now, it doesn’t match the standards and is sometimes hard to follow

The authors should also include a paragraph or two on drugs prescribed to IBD patients and how those therapy influence osteoporosis outcome if any.

Minor

Table 1 described intake of macronutrients but it fails to provide if it is mg/kg or gm/kg

The font size is not uniform throughout the manuscript. Example, line 137 – 140 and line 216 - 224 is different than entire manuscript.

Vitamin D should be included under vitamin sections

Round 2

Reviewer 1 Report

Authors notes are not attached.

Changes to text where I indicated grammar problems were updated, but the rest of the text has not been corrected. It appears that the authors completely disregarded my advice to hire a professional language editor. This manuscript is rejected!

///

On the Review Report Form/Authors' Responses to Reviewer's Comments/Author's Notes File, I have now read the Report Notes file submitted by the reviewers. The authors report that they made the necessary translations and proofreading corrections. However, when I reviewed their revised manuscript, corrections were only made to about the first third of the manuscript which had previously commented on. The rest of manuscript remains uncorrected.

Reviewer 2 Report

Although the authors have improved and addressed some of the concerns about the manuscript detailing the role of nutrients in the prevention of osteoporosis in IBD patients, there are still major grammatical and English editing changes required throughout the manuscript and as of now still doesn’t match the standards.